# Determinants of Corporate Anti-Corruption Disclosure: The Case of the Emerging Economics

**Maider Aldaz Odriozola \*** and **Igor Álvarez Etxeberria**

Faculty of Economics and business, Section of Gipuzkoa, University of the Basque Country, 20018 Donostia, Spain; igor.alvarez@ehu.eus
\* Correspondence: maider.aldaz@ehu.eus

**Abstract:** Corruption is a key factor that affects countries' development, with emerging countries being a geographical area in which it tends to generate greater negative effects. However, few empirical studies analyze corruption from the point of view of disclosure by companies in this relevant geographical area. Based on a regression analysis using data from the 96 large companies from 15 emerging countries included in the 2016 International Transparency Report, this paper seeks to understand what determinants affect such disclosure. In that context, this paper provides empirical evidence to understand the factors that influence reporting on anti-corruption mechanisms in an area of high economic importance that has been little studied to date, pointing to the positive effect of press freedom in a country where the company is located and with the industry being the unique control variable that strengthens this relationship.

**Keywords:** anti-corruption reporting; quality of disclosure; emerging countries; press freedom





## 1. Introduction

The aim of this study is to extend prior research on social disclosure by analyzing the determinants of corporate anti-corruption disclosure toward emerging economies. One of the main global problems that endanger the development of countries worldwide is corruption [1–3]. The European Union [4] (p. 3) pointed out that "corruption alone is estimated to cost the EU economy EUR 120 billion per year, just a little less than the annual budget of the European Union". The World Bank also estimated that worldwide bribery costs at least USD 1 trillion a year [5]. According to Hess and Ford [6], although international standards have been issued to fight corruption, "corporations' payment of bribes continues as a common business practice" (p.312).

This worrying situation has attracted the attention of corporations, accounting standard bodies, and accounting scholars. Nowadays, anti-corruption practices are becoming a fundamental part of companies' sustainability reporting [7]. At the same time, indicators of sustainability reporting standards (e.g., the Global Reporting Initiative) are concerned with corporate anti-corruption practices [5], which has led to a greater number of indicators in sustainability report proposals, at least in terms of quantity [8]. It should be noted that the disclosure of corruption-related information by companies can occur in a variety of ways, such as corporate codes of conduct, corporate websites, etc., and not only through sustainability reports, which is the source of information on which our study is mainly based. Links between the corruption of countries and accounting issues, including reporting, have also been analyzed within the accounting field of research. Malagueño et al. [9] discovered that the quality of accounting practices and the perceived quality of auditing systems are related to less perceived corruption at the country level. Wu [10] found a negative relationship between better accounting practices and the amount of bribe payments in 12 Asian countries. Houque and Monem [11] found that "low corruption is positively related to the length of IFRS "The International Financial Reporting Standards" experience and to the extent of disclosure" (p. 376). Blanc et al. [12] demonstrated that country-level

press freedom is crucial to explain differences in anti-corruption disclosure among the largest multinational companies. Barkemeyer et al. [5] found that companies more exposed to corruption tend to report less information about anti-corruption engagement, and Alvarez Etxeberria and Aldaz Odriozola [1], showed that, among the largest European companies, anti-corruption disclosures correlate positively with companies' reputation. Mazzi et al. [13], analyzing a sample of European companies, found that the level of compliance with mandatory disclosure requirements is related to the level of corruption, and Xu et al. [14] found a positive relationship between regional anti-corruption polices in China and companies' Corporate Social Responsibility (CSR) reporting. Branco et al. [15] found that publicly listed companies and United Nations Global Compact (UNGC) members disclose more than their counterparts. However, Sari et al. [16] found that UNGC membership is not a significant determinant, but rather, they identified companies' dependence on government tenders and foreign ownership as significant variables. Nevertheless, the analysis of accounting and corruption, particularly in the sustainability reporting area in emerging countries, is still lacking in the literature.

Traditionally, sustainability reporting literature has focused on companies from the US and from European countries. However, this situation is changing, and nowadays, emerging economies have become a popular scenario to analyze corporate reporting [17,18]. Emerging countries encompass nearly three quarters of the world's land mass, and furthermore, they are growing at a faster rate than developed countries, with China and India leading this process [19], which has led to the rise in emerging economies [18]. In this sense, the Human Development Report of the United Nations Development Programme [20] states that "the combined economic output of three leading developing countries alone—Brazil, China and India—will surpass the aggregate production of Canada, France, Germany, Italy, the United Kingdom and the United States" (p. 6). This situation has led to an increase in investment portfolios, focusing on shares from emerging market companies. Therefore, the demand for information on corporate accountability has also increased [21]. Thus, it is necessary to understand how CSR manifests itself in developing countries [22], including information about the quality of anti-corruption programs [23–26]. However, very little is known about the corporate anti-corruption disclosure phenomenon in this context. Azizul Islam et al. [27] have conducted one of the few studies on anti-corruption disclosure in emerging countries. They analyzed two major Chinese telecommunications corporations from 1995 to 2010, finding that the level of disclosure on corruption is related to international concerns over bribery practices at the country level, concluding that the main reason that those companies increased their disclosure is to gain trust from their global stakeholders. Paradoxically, even corruption is considered a key societal aspect with a negative influence, in particular, on emerging countries' economic development, and contrary to what occurs in other environments that are less affected by this social problem, the determinants of corporate anti-corruption reporting quality in emerging countries have not been analyzed in depth.

Therefore, the aim of this paper revolves around the analysis of corporate anti-corruption disclosure-quality determinants in the context of emerging economies. In our study, in order to analyze the relationship between the quality of anti-corruption information and press freedom, we consider financial variables (size and profitability), variables related to market exposure (listed companies or not), as well as industry characteristics and their sensitivity to corrupt practices. With these variables, we construct two models: the first one refers to the field and the second to the field and organization. In our study, we found that our dependent variable press freedom is positively and significantly correlated with anti-corruption reporting and that the industries' corruption sensitivity has a positive impact on this relationship.

In this sense, our paper contributes to the literature in several ways. On the one hand, our study provides empirical evidence to the numerous papers published that attempt to analyze the factors conditioning companies' disclosure in terms of anti-corruption procedures. Most of the studies focus on the behavior of companies in Western countries

and large economies, whereas our paper aims to study the reality of this disclosure in a little studied context of great importance, both for its growing weight in the global economy and the greater influence that corruption has on their economies: emerging countries. The results obtained have important implications for understanding companies' behavior in their reporting, as the literature has had little impact through academic work in the area of emerging countries. Furthermore, the results can help investment portfolio analysts when evaluating their investments, as they have increasingly larger positions in this area of great economic potential. The last contribution derives from the measurement of the quality of reporting on corruption; our paper establishes a stricter criterion than those previously used in the literature [5,12,26–28]. All these papers carry out a content analysis based on various proposals and evaluate the amount of information provided without including any aspect to assess the quality of such information. We consider that the proposed model for assessing the quality of information is another contribution to the evaluation of companies' social behavior, as we do not rely solely on the disclosure of corruption-related aspects, but rather, we weight this information in terms of the quality of the information provided. It is assumed that the quality of information will be higher when companies report on all possible aspects and not only on those aspects that may be in their interest to report; therefore, only those that report on the three aspects under analysis will obtain a positive score.

This paper is structured as follows: In the following section, we introduce anti-corruption disclosure and the review-related research and develop our working hypotheses. Next, we present the methodology we used for our empirical analysis, and we outline the results. Finally, we conclude with a discussion and conclusion section underlining some implications.

## 2. Anti-Corruption Reporting: Driving Forces in Emerging Countries

CSR disclosure research is biased toward corporate environmental reporting. Dozens of studies can be found regarding the quantity and quality of the environmental information disclosed by companies and its relationship with the industry to which they belong [29]; the size of the company [30]; the media exposure [31]; the companies' environmental performance [32]; and many other variables at the individual, organizational, and socioeconomic levels [33]. However, since the beginning of the research on social and environmental accounting, the researchers have also been interested in the social information disclosed by companies. Among the pioneering papers in the area of social reporting, we can find the work of Adams [34], who analyzed corporate equal opportunities reporting in Britain. Adams found that most companies' disclosures in this area are due to compliance with law requirements and that "only a minority of companies comply fully with the legislation" (p. 87). Gray et al. [35] studied social and environmental corporate reporting in the annual reports of UK companies over a period of 13 years. They show that nonenvironmental matters such as employee-related issues and information about the community (mainly charitable donations) were the most popular topics reported from 1979 to 1991. Grosser and Moon [36], analyzing gender equality reporting among UK companies, found that, even though the quality of the information has improved in some dimensions, "accountability in the provision of gender information is regarded in terms of responsiveness to changing expectations of company behavior by market actors" (p. 194). Taggesons et al. [37] found that some industries are more active with internet-based ethical disclosure than other industries in the Swedish context.

However, even though corporate social performance is crucial to understand the CSR phenomenon, this dimension of sustainability is much less studied [38], attracting less attention from researchers in this area than the environmental side.

"Using accounting to link corporate action with its social impact and implications is challenging, and this challenge, combined with the relative lack of attention that has been given to social sustainability, means that there has not been much written in this area" [39] (p. 160).

Little is known about corporate social disclosure in general but even less is known about anti-corruption reporting. For example, Blanc et al. [40] found that Siemens strategically increased the information reported on corruption in order to reduce the negative impact on its legitimating of a corruption scandal. Azizul Islam et al. [41] showed that anti-bribery disclosure is associated with the activities of the media and NGOs, whereby this information has a symbolic content more closely linked to the aim of increasing their legitimation than their accountability practices in the telecommunications sector. Alvarez Etxeberria and Aldaz Odriozola [1] showed that anti-corruption disclosure among the 70 largest European companies correlates with corporate social reputation. Blanc et al. [28], when analyzing a sample of 105 largest multinational companies, also found that corporate anti-corruption disclosures are related to media exposure and a country's press freedom.

When the literature aims to analyze the quality of the information, different models have been used. For example, Blanc et al. [12,28] used an index of information on anti-corruption proposed by TI [42], which considers 13 items. Specifically, they calculated the percentage of information provided (0 if they do not report, 0.5 if they report partially, and 1 if they provide complete information) with respect to the maximum possible score (13 points = 100%). Saenz and Brown [26] and Azizul Islam et al. [27] developed their own measurement instruments based on one of the various international anti-bribery guidelines. The first identified five general areas: (1) accounting for combating bribery, (2) board and senior management responsibility, (3) building human resources to combat bribery, (4) responsible business relations, and (5) external verification and assurance. They then counted the number of topics covered in their annual reports and CSR reports. In contrast, in the second, four topics were identified: (1) leadership and commitment of senior management, (2) control and evaluation, (3) planning, and (4) implementation. Then, the information in each of the items included within each theme was evaluated on a scale of 0 to 3, subsequently giving a different weight to each indicator depending on the number of indicators that make up each general theme, analyzing the annual report, corporate social responsibility report, and code of ethics/conduct. Finally, Barkemeyer et al. [5] used the number of Global Reporting Initiative (GRI) G3 reports published by the companies and examined the official information disclosed by the companies in their social responsibility reports and in various documents published on their corporate websites regarding their anti-corruption actions. In our case, we based our paper on the information published by companies on their websites or relevant links to measure whether they report on three aspects related to anti-corruption practices: (1) their anti-corruption programs, (2) their company structures and holdings, and (3) country-by-country information on key financial aspects. We consider that the reported data are of better quality if companies report on all three aspects.

The majority of previous studies focused their attention on major economies; however, corruption is especially concerning among poor and emerging countries [43]. In this respect, the papers on emerging countries focus on a small number of companies in a single country [26] or, if they do analyze a more important sample of companies and countries, they are descriptive, not explanatory, studies [5]. Among these, we could highlight papers such as that by Saenz and Brown [26], who analyzed the disclosure of corruption measures of a sample, which included 10 companies operating in Latin America, finding that information about the leadership and commitment of senior management and the control and evaluation were the most popular. Barkemeyer et al. [5] showed country- and sector-level differences in anti-corruption disclosing patterns among a sample of 933 sustainability reports. According to these authors, contrary to Eastern European companies, South and East Asian companies show particularly "high levels of coverage of GRI indicators about corruption" (p. 363). However, very little is known about the particularities of corporate anti-corruption disclosure phenomenon in this context. For Belal et al. [17], emerging countries are based on different social, cultural, and politic institutions and, therefore, the different structures and business cultures of companies from those countries require further understanding of their social activities. In this sense, our

paper aims to understand the reasons behind the differences in the application of corruption processes of companies belonging to emerging countries, and in order to understand those differences, we rely on the influence of press freedom as an independent variable.

As the main objective of our study is to understand the relationship between press freedom and anti-corruption mechanisms, and due to the lack of a theoretical framework to help us understand this relationship, in this paper, we considered it appropriate to study the literature that relates these two concepts from a macro perspective. In contrast to the business-related literature, the negative relationship between press freedom and corruption [2] and between the country's level of corruption and companies' transparency [13,44] is very well-documented. Press freedom is considered as an external control of corruption [3]. Based on the theoretical framework provided by *Political Economics* [45,46] (p. 94), we can highlight the following models:

Press freedom → Democracy → State of corruption

Increased press freedom is seen as an institutional factor that enhances the democracy of societies, which in turn has a positive effect on reducing corruption in those societies. In that sense, Freille et al. [47] found that "economic and political influence on the media" has an impact on corruption. Therefore, from the macro perspective, it is assumed that press freedom contributes to the democratization processes in countries [3], strengthening the accountability and transparency of the government institutions [48] and, in the same way, decreasing information asymmetries [3]. The increase in accountability due to press freedom in societies is a key factor in reducing corrupt actions, a fact that has been widely probed in academia [2,3,47], even when controlling by other key factors of corruption [47]. Bhattacharyya and Hodler [49] found a complementary effect of democratization and media freedom on corruption, analyzing a sample of 129 countries from 1980 to 2007.

In this paper, we assume that this process could be replicated at the micro level, i.e., in the area of companies. The institutional approach of a society could shape companies' performance [50]. Based on this line of argument, we consider that societies with a higher level of accountability/transparency—taking the press freedom institution as a variable that allows us to measure and rate the degree of accountability/transparency of society—are more concerned about combating corruption and could condition the companies' behavior and, more specifically, their accountability processes. The concept of accountability is closely related to a company's disclosure. Therefore, we can expect a better quality of social disclosure and, more specifically, of this information related to their anti-corruption processes.

Therefore, following previous literature about the importance of press freedom, quality of democracy, and corruption and corporate anti-corruption reporting practices and the current poor understanding of this reporting behavior within emerging countries, this will be the hypothesis to be tested in this paper.

**Hypothesis 1 (H1).** *Country-level press freedom will affect the quality of corporate anti-corruption reporting of multinational companies in emerging countries.*

## 3. Materials and Methods

Our sample is composed of 95 companies from emerging countries (34 from China; 19 from India; 10 from Brazil; 6 from Mexico; 5 from Russia; 4 from Thailand and South Africa; 3 from United Arab Emirates; 2 from Chile, Indonesia, and Turkey; and 1 from Argentina, Egypt, Hungary, and Malaysia). Their mean size (based on FY 2015) is EUR 53.2 million (USD 19,762.5 million), although there is a great diversity in the sizes in the sample ranging from USD 404.5 million to USD 137,909.3 million. This sample is based on "Transparency in Corporate Reporting. Assessing Emerging Market Multinationals", a report published by Transparency International (TI), the global coalition against corruption, in 2016. This report measures companies' reporting of information in three areas: "the reporting of key elements of their anti-corruption programs, the reporting of their company structures and holdings, and the reporting of key financial information on a country-by-

country basis" [51] (p. 6). The original report analyses the situation of 100 companies, but five of them have been excluded from our sample due to lack of information. Thus, we do not collect dependent variable data directly, but we relied on the data provided by the nongovernmental organization in the report and on consultations made with them to clarify the details of the methodology they used. Note that the study considered data collected by a researcher from TI through desk research between November 2015 and January 2016 using the latest available documentation on company websites or relevant links; thus, our study covers the period between 2014 and 2015. Finally, all data points collected were independently validated by a second researcher from the same organization. Table 1 shows the formulas used to calculate all the variables of our models.

**Table 1.** Formulas used to calculate dependent, independent, and control variables.

| Type | Name | Source | Formula |
|---|---|---|---|
| Dependent variable | Anti-Corruption Reporting (ACR) | "Transparency in Corporate Reporting. Assessing Emerging Market Multinationals" report, published by Transparency International (TI) * | Dummy variable IF (ACP × OT × CBC > 0; ACR = 1) IF (ACP × OT × CBC = 0; ACR = 0) |
| Independent variable | Press Freedom (Pressfre) | Press Freedom Index 2016 report, published by Reporters Without Borders | (Score offered by the organization) ×-1 |
| Control variables | Industry Risk | Bribe Payers Index 2011, published by Transparency International (TI) | Dummy variable 1 if industry below the mean 0 if industry above the mean |
| | Public Listed | ORBIS database | Dummy variable |
| | Firm size | | Natural log of the mean total assets of 2014–2015 in USD |
| | Financial Performance | | Mean of ROA of 2014–2015 |

* Score offered by the organization composed of three categories: Anti-Corruption Programs (ACP), Organizational Transparency (OT), and Country-By-Country reporting (CBC), ORBIS database, with economic and financial information on more than 200 million privately held companies worldwide.

The empirical model that we developed aims to analyze whether the information on anti-corruption practices reported by companies is influenced by the outside perception of the level of corruption suffered by the companies' countries of origin. The dependent variable in our model is therefore the anti-corruption reporting score, and we use press freedom to measure the level of transparency and accountability of the emerging countries where the companies are located. First, we present a model where only the characteristics of the company's area of business are considered; thus, as control variables, we include industry risk and stock market participation. In addition to this model, we include a second model where the company's characteristics are also considered; thus, we include another two variables as control variables: the company's size and financial performance:

Model 1: Field

$$ACR_{i,t} = \beta_0 + \beta_1\,Presfre_{i,t} + \beta_2 IndRisk_{i,t} + \beta_3 Publiclisted_{i,t} + \varepsilon \tag{1}$$

Model 2: Field + organization

$$ACR_{i,t} = \beta_0 + \beta_1\,Presfre_{i,t} + \beta_2 IndRisk_{i,t} + \beta_3 Publiclisted_{i,t} + \beta_4\,Firm\,Size_{i,t} + \beta_5 ROA_{i,t} + \varepsilon \tag{2}$$

Dependent variable: ACR (anti-corruption reporting).

The "Transparency in Corporate Reporting. Assessing Emerging Market Multinationals" report, published by TI, provides an overall score of emerging market multinationals related to their reporting practices. The categories reporting on anti-corruption programs,

organizational transparency, and country-by-country reporting are assessed to obtain the overall assessment. The first category analyses if companies have good governance practices in place, meaning they act as responsible corporate citizens. The second category analyses if the disclosure made by sample companies provides enough information to their stakeholders to help them identify possible fraudulent actions around the whole perimeter of the organization. The last category analyses if the company reports allow their stakeholders to evaluate the impact of their business in all the countries where they operate.

The first category clearly relates to companies' transparency regarding their anti-corruption practices, as the other two dimensions relate to companies' general transparency. Given these metrics, we consider that the companies reporting on the three dimensions have more credibility in all the aspects on which they report and, therefore, in relation to their anti-corruption program dimensions. Based on Archel and Larrinaga [52], we consider that the companies disclosing their CSR activities, bearing in mind wider organization boundaries and considering only the parent company instead, are more demanding regarding the disclosure made to their stakeholders, making it easier for those stakeholders to interpret the information reported in the CSR area. With this in mind, instead of adding the scores of the three dimensions (as originally carried out by TI in their report), we multiply them. Therefore, we increase the requirement to classify a company as transparent on their anti-corruption practices. It is not sufficient to report on the dimension of anti-corruption programs. They must also report on the other two dimensions; otherwise, their final score will be "0".

By applying this score, our purpose is to evaluate not only the quantity of disclosure of anti-corruption programs but also the quality. This method is innovative, as it is more restrictive than others used in the literature, which are only based on the quantity of anti-corruption reporting using the mean of those three values [12,28] or the quantity of information reported measured using other systems [5,26,27].

Once we multiply the scores of the three dimensions, we obtain our anti-corruption reporting score; for the sample companies, this score ranged from 0% to 30%, with a mean (median) of 4% (0%). We then transform it into a dummy variable, 1 if in the previous step the company has a score greater than 0% and 0 otherwise.

### 3.1. Independent Variable: Press Freedom

We rely on the Press Freedom Index 2016 developed by the nongovernmental organization Reporters Without Borders (RWB). The index reflects whether, during 2015, media independence in the sample countries suffered any attacks, regardless of whether the origin of those attacks was the government or the private sector.

RWB uses six criteria to determine the index; these criteria—pluralism, media independence, media environment and self-censorship, legislative framework, transparency, and the quality of the infrastructure that supports the production of news and information—are evaluated through a questionnaire given to several experts. The index also takes into consideration the physical attacks suffered by journalists. RWB provides a ranking of 180 different countries in terms of press freedom (1 represents the country with the highest press freedom and 180 represents the country with the lowest).

We multiplied the score offered by the organization by −1 so we obtain a ranking where countries with greater press freedom obtains higher scores in order to facilitate interpretation.

### 3.2. Control Variables: Size, Financial Performance, Industry Risk, and Publicly Listed

First, we included the organizational size because, from a cost perspective, this may have an effect on business communication [31,53,54], as the costs arising from the publication of data are easier to absorb by large companies. In this sense, most studies in the area show a positive relationship between organizational size and the quantity of social and environmental reporting [31,53–55]. At the same time, larger companies are thought to have greater financial slack to support investment related to social responsibility [56]. Thus,

we can expect that size will positively influence the relationship studied in the paper. We measure companies' size using a natural log of the mean total assets of 2014–2015 in USD.

Second, we include financial performance in our model; a variable that has been widely used in this context [57,58]. The disclosure made by companies is a way to establish and maintain a dialogue between the companies and their stakeholders, aiming to survive in an uncertain and competitive environment [21]. In this sense, those companies with higher profitability would want to make their situation public [59]. Thus, although the relationship between social performance and financial performance is not yet clear [27,59], we expect a positive relationship between profitability and reporting. We used the mean of ROA (Return on Assets)—which is calculated by dividing earnings before interest and tax (EBIT) by the mean total assets [55,60]—of 2014 and 2015 to measure financial performance.

Third, industrial risk was considered, bearing in mind that the companies' sector may influence the quantity of information related to social and environmental practices reported by those companies [29,53]. In particular, more recent papers considering sector as a reporting quantity explanatory variable focus on the sensitivity of those sectors to social and environmental issues [55,61]. To consider the sensitivity of different sectors on corruption issues, we use TI's [62] Bribe Payers Index of industry sectors. We then define a binary variable, coded as "1" if the company's primary industry is rated below the mean and as "0" if the company's primary industry is rated above the mean. Thus, we consider oil and gas, and utilities as corruption-sensitive industries while the other industries in the sample were taken to be non-corruption sensitive.

Finally, publicly listed companies were included in the model. In the literature, disclosure made by companies was seen as an essential tool for a stock market to work efficiently [63] and to reduce information asymmetry between managers and investors [64]. Haniffa and Cooke [65] stated that regulations developed by most of the stock exchange markets around the world lead companies to improve their social and environmental disclosure. At the same time, pressure made by different interest groups to disclose on different aspects of business activities seems to be greater for publicly listed companies because of contractual and legitimacy processes. Companies that are fully controlled by the participants in capital markets face high potential political costs and are highly visible targets [66]. Therefore, we expect publicly listed companies to report more on CSR aspects. This is also a binary variable coded as "1" if the company is publicly listed and "0" otherwise.

Due to the nature of the variables analyzed, we use the Spearman correlation method to analyze the correlation among the variables used in the study. Furthermore, considering the nature of our dependent variable, we run a logit regression to analyze whether the quality of anti-corruption reporting among the sample companies is influenced by press freedom.

First, we control for variables at the sector level (the corruption sensitivity of the industry where the company operates and whether it belongs to a publicly traded stock market); thus, we apply the logit regression to our first model (field):

$$\text{Pr (ACR} = 1/\text{Presfre, IndRisk, Publiclisted)} = F\ (\beta 0 + \beta 1\ \text{Presfre}_i + \beta 2\text{IndRisk}_i + \beta 3\text{Publiclisted}_{di} + \beta 4\ \text{Firm Size}_i + \beta 5\text{ROA}_i) \tag{3}$$

Subsequently we control for variables at the organizational level (size and profitability); thus, we apply the logit regression to our second model (field + organization):

$$\text{Pr (ACR} = 1/\text{Presfre, IndRisk, Publiclisted, Firm Size, ROA)} = F\ (\beta 0 + \beta 1\ \text{Presfre}_i + \beta 2\text{IndRisk}_i + \beta 3\text{Publiclisted}_{di} + \beta 4\ \text{Firm Size}_i + \beta 5\text{ROA}_i) \tag{4}$$

## 4. Results

Table 2 presents descriptive statistics for every single variable used in this study (Appendix A shows all the disaggregated data used for the companies in the sample). Most of the companies analyzed are large, listed (73%), and private companies (74%). Almost

40% of companies belong to sectors "sensitive" to corruption practices, and on average, the quality of the reporting on anti-corruption practices is low. It can be also observed that press freedom in the countries where sample companies are located is quite low, but our data show great differences between countries. Finally, the difference in the profitability of the sample companies is also quite large.

**Table 2.** Descriptive statistics of variables analyzed.

| Variable | Mean | SD | Min | Max |
|---|---|---|---|---|
| ACR | 0.400 | 0.493 | 0 | 1 |
| Presfre | −139.915 | 38.354 | −31 | −176 |
| Public List | 0.737 | 0.443 | 0 | 1 |
| Size | 6.998 | 0.477 | 6.120 | 8.160 |
| ROA | 5.881 | 11.676 | −34.010 | 47.610 |
| IndRisk | 0.389 | 0.490 | 0 | 1 |

Note. ACR = Anticorruption reporting score, Presfre = Press Freedom Index 2016, Public list = whether a company is publicly listed or not, Size = Firm's size measured by natural log of the mean total assets of 2014–2015 in USD, ROA = financial performance measured by the mean ROA (Return on Assets) of 2014 and 2015, and Industry Risk = sensitivity of the sector to corruption practices.

The correlation results (see Table 3) show that our dependent variable press freedom is positively and significantly correlated with anti-corruption reporting. Regarding the control variables, we observe that being publicly listed is positively and significantly correlated with our dependent variable. On the other hand, none of the other control variables—size, ROA, and industry risk—appear to be significantly correlated with the quality of corporate anti-corruption disclosure.

**Table 3.** Spearman's Rho nonparametric correlation coefficients.

| | ACR | Presfre | Public List | Size | ROA | Ind. Risk |
|---|---|---|---|---|---|---|
| ACR | 1 | | | | | |
| Presfre | 0.520 ** | 1 | | | | |
| Public List | 0.390 ** | 0.435 ** | 1 | | | |
| Size | −0.117 | −0.143 | −0.111 | 1 | | |
| ROA | 0.111 | 0.101 | 0.020 | −0.047 | 1 | |
| IndRisk | 0.141 | −0.043 | −0.111 | −0.029 | −0.290 * | 1 |

Significant levels (based on a two-tailed test). ** The correlation is significant at level 0.01. * The correlation is significant at level 0.05.

The logit regression results (Table 4) show a positive (0.016) and significant ($p = 0.039$) relationship between press freedom and the quality of corporate anti-corruption disclosure. This result is in line with Blanc et al. [15,36]. Therefore, we accept the hypotheses proposed.

**Table 4.** Results of the logit regression analysis.

| Variable | Model 1 (Field) | | | Model 2 (Field + Organization) | | |
|---|---|---|---|---|---|---|
| | Coef | Wald | Sig | Coef | Wald | Sig |
| Constant | | | | | | |
| Presfre | 0.023 ** | 8.041 | 0.005 | 0.016 * | 4.250 | 0.039 |
| Public List | 2.131 * | 6.655 | 0.010 | 1.773 | 2.750 | 0.097 |
| IndRisk | 1.051 * | 4.037 | 0.045 | 1.519 * | 5.444 | 0.020 |
| Size | | | | −0.454 | 0.655 | 0.418 |
| ROA | | | | 0.029 | 1.426 | 0.232 |
| R Square | | 31.8% | | | 29.7% | |
| N = 95 | | | | | | |

Significance levels (based on a two-tailed test). ** The correlation is significant at level 0.01. * The correlation is significant at level 0.05.

On the other hand, regarding field level variables and, specifically, the industries' sensitivity to corruption (Ind. Risk), we contrast previous literature [1,12,28] and our results show a significant relationship with the quality of the anti-corruption reporting. Whilst the papers by Alvarez Etxeberria and Aldaz Odriozola [1] and Blanc et al. [28] find that companies operating in more sensitive industries report less about their anti-corruption practices, Blanc et al. [12] and our results reflect the contrary. Our results also reflect a nonsignificant relation between being a publicly listed company (Public List) and the quality of anti-corruption reporting practices.

Finally, focusing on organization level variables, size appears to be a nonsignificant variable in our study, in line with most previous literature [1,12,28]; however, Xu et al. [14] indicated a positive and significant effect of the size of the companies on their disclosure levels. For profitability, the results are also conflicting. Based on our analysis, profitability is not a significant variable, and similar results are shown in Blanc et al. [28]; however, Alvarez Etxeberria and Aldaz Odriozola [1] found a positive and significant relationship between those variables.

The low alterability of the results of both models—model 1 (field) or model 2 (field + organization)—is proof of the robustness of the results. Nevertheless, for more evidence of the robustness, we ran a lineal regression. The results of the lineal regression analysis (Table 5) are similar to the results obtained on the logit regression analysis.

**Table 5.** Results of the lineal regression analysis.

| Variable | Model 1 (Field) | | | Model 2 (Field + Organization) | | |
|---|---|---|---|---|---|---|
| | Coef | t-statistic | Sig | Coef | t-statistic | Sig |
| Constant | | | | | | |
| Presfre | 0.004 ** | 3.477 | 0.001 | 0.003 * | 2.215 | 0.030 |
| Public List | 0.317 ** | 2.955 | 0.004 | 0.310 | 1.804 | 0.076 |
| IndRisk | 0.190 * | 2.111 | 0.037 | 0.293 * | 2.429 | 0.018 |
| Size | | | | −0.100 | −0.875 | 0.385 |
| ROA | | | | 0.006 | 1.111 | 0.271 |
| R Square N = 95 | | 28.2% | | | 22.3% | |

Significance levels (based on a two-tailed test). ** The correlation is significant at level 0.01. * The correlation is significant at level 0.05.

The only difference between the "field"-related model and the "field and organization"-related model in both analyses (logit regression and lineal regression) is in the variable that refers to participation in a public capital market. This variable seems to be significant at the field level but not at the organizational level. This could be because we do not differentiate between companies that participate in foreign capital markets and those that only participate in domestic markets. As Haniffa and Cooke [65] pointed out, companies listed on the domestic capital market in a developing country will be less prone to disclosure because of the lack of regulation and because they would suffer less pressure from their stakeholders. On the contrary, if a company is part of a foreign capital market, where regulation and pressure is higher, it will increase the quantity and the quality of its disclosure.

## 5. Discussion and Conclusions

The aim of this paper is to analyze two aspects that have become hot topics in recent decades in the global economic and social field: corporate corruption and emerging countries. Although we can find plenty of studies that analyze companies' social and environmental reporting in order to understand their CSR performance, their social dimension is clearly under-studied [38,39], and in the case of their corruption disclosure, this is even more pronounced [1]. Parallel to this gap, the relevance of emerging countries in world economic development is clearer year by year. As a result, it becomes a relevant scenario to analyze and understand the companies' performance and corruption disclosure, especially

when corruption is accepted as a one of the main problems that could hinder development of those regions.

Specifically, this paper fills this gap in order to understand which determinants influence the quality of a company's anti-corruption disclosure in emerging countries. Therefore, in line with other authors [3,47,49], we consider that press freedom could be a key factor-independent variable that could influence companies' disclosure on corruption. The literature on political economy argues that press freedom is a relevant institutional factor in reducing corruption in societies. Based on these studies, our hypothesis aims to assess whether this process also occurs at the level of the business in the little-studied area of emerging countries. With this aim in mind, we developed two models where the dependent variable is the anti-corruption reporting score and where press freedom (independent variable) refers to the level of accountability/transparency of the emerging countries where the companies are located. Besides trying to develop a more robust understanding of this relationship, we established two different groups of control variables related to the organizational level (size and profitability) and the sector level (industry risk and publicly listed). Our empirical texts show different and interesting findings that allow us to understand this relevant relationship in companies from emerging countries.

Firstly, in the first correlation (Spearman), where all control variables are used together, the results show that press freedom is positively and significantly correlated with anti-corruption reporting, so we can accept the hypothesis. This finding clearly contributes to understanding the behavior of global companies located in emerging countries. These results reinforce the model employed at the macro level or at the micro or company levels. We believe that the finding that this relationship is positive in companies belonging to emerging countries helps us to understand how the institutional framework of the country conditions the companies' degree of accountability. From the country perspective, we demonstrate that institutional characteristics affect companies' behavior and, in particular, the country's level of transparency. From the political-economic perspective, Chowdhury [46] assumes that press freedom generates a positive effect on the transparency of the society and therefore reduces information asymmetry [3], strengthening the accountability of governments [48]. Our study shows how those macro effects could be translated to the business field. In those countries where accountability is empowered by press freedom, companies could also be conditioned to improve their accountability processes, as this improves their disclosure on what tools are implemented to avoid corrupt behavior. These findings provide interesting information for global investment portfolio managers when including risk assessment in their evaluation and decision-making processes.

Secondly, at the organizational level, the results show that the companies' size and profitability are not aspects that influence the quality of anti-corruption disclosure. However, in terms of understanding those nonsignificant relationships, we must consider that all companies analyzed are large. Therefore, we can assume that, when companies are large, their financial resources are not a significant aspect in discriminating them based on their disclosure of social information and, in this specific case, of information regarding corporate anti-corruption practices.

Thirdly, the sector level field is a key factor in understanding the implementation of the disclosure of anti-corruption policy information in our study. On the one hand, there are very few empirical studies in the literature focusing on sector-level analysis of emerging countries. In this sense, Barkemeyer et al. [5] found relatively few differences in the sector-level, but they measured corporate social responsibility disclosure in general, not only anti-corruption-related information, whereas Azizul Islam et al. [27] only focused on the industry to understand their specific motivation for social disclosure. On the other hand, it shows that being part of a corruption-sensitive industry in an emerging country is a determining factor in increasing the quality of the anti-corruption disclosure.

We believe that this paper contributes to a better understanding of the anti-corruption behavior of companies in a very relevant geographical area such as emerging countries. In that sense, we consider that, for professionals, particularly those related to portfolio

management and risk assessment focused on emerging countries, the fact that those countries where the country's transparency positively affects the quality of information on corruption is a highly relevant aspect to be considered.

We would also like to point out the limitations of our study. It must be noted that corruption can be measured very differently; thus, the different regulations of the sample countries make some countries much more demanding in corruption-related matters, and this aspect could affect our results. In turn, we have left out of our analysis the discussion on the alleged benefits of a certain level of corruption. Finally, the trust that a country's inhabitants place in its media can influence how press freedom affects the transparency of companies in that country.

Therefore, the abovementioned limitations open future lines of research; we also consider that an analysis of the differences between countries could provide more and better knowledge as well as could further analyze the implications of our findings and the financial, economic, and social behaviors of these companies.

**Author Contributions:** All the authors contributed to conceptualization, formal analysis, investigation, methodology, and writing and editing of the original draft. All authors have read and agreed to the published version of the manuscript.

**Funding:** This research was funded by the Spanish Ministry of Science, Innovation, and Universities and by the Basque Government (grant numbers PID2019-107822RB-I00 and IT1073-16).

**Institutional Review Board Statement:** Not applicable.

**Informed Consent Statement:** Not applicable.

**Data Availability Statement:** The data will be made available upon request to the corresponding author.

**Conflicts of Interest:** The authors declare no conflict of interest.

## Appendix A

| Company | Country | Sector | TI Company Index 2016 | Anti-corruption programs ACP | Organizational transparency OT | Country-by-country reporting CBC | ACP*OT*CBC | ACR | Independent (Press Freedom) | Industry risk | Public listed(yes=1) | Firm Size (log media AT2014_2015) | ROA (media 2014-2015) |
|---|---|---|---|---|---|---|---|---|---|---|---|---|---|
| ALIBABA GROUP HOLDING LIMITED | China | Consumer services | 2.5 | 42% | 31% | 0% | 0% | 0 | -176 | 0 | 1 | 7.690 | 14.531 |
| ALUMINUM CORPORATION OF CHINA LIMITED | China | Basic materials | 1.5 | 27% | 19% | 0% | 0% | 0 | -176 | 1 | 0 | 7.485 | -4.164 |
| ANSHAN IRON & STEEL GROUP CORPORATION | China | Basic materials | 0.8 | 23% | 0% | 0% | 0% | 0 | -176 | 1 | 0 | | |
| BAJAJ AUTO LTD. | India | Consumer goods | 4.7 | 19% | 88% | 33% | 6% | 1 | -133 | 0 | 1 | 6.400 | 20.994 |
| BAOSTEEL (ZHANJIANG) GROUP CO., LTD. | China | Basic materials | 2 | 23% | 38% | 0% | 0% | 0 | -176 | 1 | 0 | 6.838 | |
| BHARAT FORGE LIMITED | India | Industrials | 4.1 | 19% | 75% | 30% | 4% | 1 | -133 | 0 | 1 | 6.125 | 8.299 |
| BHARTI AIRTEL LIMITED | India | Telecommunication | 7.3 | 88% | 100% | 30% | 26% | 1 | -133 | 0 | 1 | 7.513 | 2.544 |
| THE BIDVEST GROUP LIMITED | South Africa | Consumer services | 3.7 | 62% | 50% | 0% | 0% | 0 | -39 | 0 | 1 | 6.875 | 6.125 |
| BRF S.A. | Brazil | Consumer goods | 4.4 | 58% | 75% | 0% | 0% | 0 | -104 | 0 | 1 | 7.078 | 6.666 |
| BUMI RESOURCES TBK, PT | Indonesia | Basic materials | 4.8 | 62% | 81% | 3% | 2% | 1 | -130 | 1 | 1 | 6.602 | -32.387 |
| BYD CO., LTD. | China | Consumer goods | 2.7 | 31% | 50% | 2% | 0% | 0 | -176 | 0 | 1 | | |
| CHAROEN POKPHAND GROUP CO LTD | Thailand | Consumer services | 0.6 | 0% | 13% | 4% | 0% | 0 | -136 | 0 | 0 | 6858 | 2.803 |
| CHERY AUTOMOBILE CO., LTD. | China | Consumer goods | 0 | 0% | 0% | 0% | 0% | 0 | -176 | 0 | 0 | 7.043 | 0.938 |
| CHINA COMMUNICATIONS CONSTRUCTION COMPANY LIMITED | China | Industrials | 3.3 | 54% | 44% | 0% | 0% | 0 | -176 | 0 | 1 | 8.033 | 2.192 |
| CHINA INTERNATIONAL MARINE CONTAINERS (GROUP) CO., LTD. | China | Industrials | 3 | 35% | 56% | 0% | 0% | 0 | -176 | 0 | 1 | 7.187 | 2.838 |
| CHINA MINMETALS CORPORATION LIMITED | China | Basic materials | 0.8 | 19% | 6% | 0% | 0% | 0 | -176 | 1 | 0 | | |
| CHINA NATIONAL CHEMICAL CORPORATION | China | Basic materials | 0.7 | 15% | 6% | 0% | 0% | 0 | -176 | 1 | 0 | 7.513 | 0.254 |
| CHINA NATIONAL OFFSHORE OIL CORP. | China | Oil, gas & energy | 1.1 | 27% | 6% | 0% | 0% | 0 | -176 | 1 | 0 | | |
| CHINA RAILWAY CONSTRUCTION CORPORATION LIMITED | China | Industrials | 2.1 | 31% | 31% | 2% | 0% | 0 | -176 | 0 | 1 | | |
| CHINA SHIPBUILDING INDUSTRY CORPORATION | China | Industrials | 0,7 | 15% | 6% | 0% | 0% | 0 | -176 | 0 | 0 | | |
| CHINA SHIPPING GROUP CO., LTD. | China | Industrials | 1,2 | 31% | 6% | 0% | 0% | 0 | -176 | 0 | 0 | | |
| CHINA STATE CONSTRUCTION ENGINEERING CORPORATION LIMITED | China | Industrials | 0,3 | 8% | 0% | 0% | 0% | 0 | -176 | 0 | 0 | | |
| CHINT GROUP CORPORATION | China | Utilities | 0,4 | 12% | 0% | 0% | 0% | 0 | -176 | 1 | 0 | | |
| COTEMINAS S.A. | Brazil | Consumer goods | 1.1 | 8% | 25% | 0% | 0% | 0 | -104 | 0 | 0 | | |
| CG POWER AND INDUSTRIAL SOLUTIONS LIMITED | India | Industrials | 4 | 15% | 75% | 28% | 3% | 1 | -133 | 0 | 1 | 6.233 | -0.820 |
| DP WORLD LTD | UAE | Industrials | 3.4 | 65% | 38% | 0% | 0% | 0 | -119 | 0 | 1 | 7.281 | 4.136 |
| DR REDDY'S LABORATORIES LIMITED | India | Health care | 5.8 | 69% | 75% | 30% | 16% | 1 | -133 | 1 | 1 | 6.490 | 10.920 |
| EL SEWEDY ELECTRIC COMPANY | Egypt | Industrials | 5.7 | 77% | 81% | 12% | 7% | 1 | -159 | 0 | 1 | 6.346 | 4.650 |
| EMBRAER - EMPRESA BRASILEIRA DE AERONAUTICA S.A. | Brazil | Industrials | 5.6 | 92% | 75% | 0% | 0% | 0 | -104 | 0 | 1 | 7.043 | 1.705 |
| EMIRATES AIRLINES | UAE | Consumer services | 3.8 | 38% | 75% | 0% | 0% | 0 | -119 | 0 | 0 | | |

**Figure A1.** *Cont.*

| | | | | | | | | | | | | |
|---|---|---|---|---|---|---|---|---|---|---|---|---|
| EMIRATES TELECOMMUNICATIONS CORPORATION PJSC | UAE | Telecommunication | 2.8 | 19% | 56% | 8% | 1% | 1 | -119 | 0 | 1 | | |
| EVRAZ GROUP SA | Russia | Basic materials | 5.2 | 85% | 63% | 10% | 5% | 1 | -148 | 1 | 1 | 6.839 | -28.240 |
| S.A.C.I. FALABELLA | Chile | Consumer Services | 6.2 | 50% | 75% | 60% | 19% | 1 | -31 | 0 | 1 | 7.261 | 4.110 |
| FOMENTO ECONOMICO MEXICANO SAB DE CV | Mexico | Consumer Goods | 4.6 | 73% | 56% | 8% | 3% | 1 | -149 | 0 | 1 | 7.076 | 5.313 |
| RICHTER GEDEON VEGYESZETI GYAR RT | Hungary | Health Care | 4.6 | 46% | 88% | 4% | 2% | 1 | -67 | 1 | 1 | 6.431 | 5.355 |
| GEELY INTERNATIONAL CORPORATION | China | Consumer Goods | 0.4 | 12% | 0% | 0% | 0% | 0 | -176 | 0 | 0 | | |
| GERDAU S.A. | Brazil | Basic Materials | 3.8 | 50% | 63% | 0% | 0% | 0 | -104 | 1 | 1 | 7.319 | -2.134 |
| GRUMA, S.A.B. DE C.V. | Mexico | Consumer Goods | 3.5 | 42% | 63% | 0% | 0% | 0 | -149 | 0 | 1 | 6.426 | 11.006 |
| GRUPO ALFA SA DE CV | Mexico | Basic Materials | 2.8 | 38% | 44% | 2% | 0% | 0 | -149 | 1 | 1 | | |
| GRUPO BIMBO, S.A.B. DE C.V. | Mexico | Consumer Goods | 3.7 | 92% | 19% | 0% | 0% | 0 | -149 | 0 | 1 | 7.073 | 2.285 |
| HINDALCO INDUSTRIES LIMITED | India | Basic Materials | 5 | 46% | 75% | 30% | 10% | 1 | -133 | 1 | 1 | 7.342 | 0.315 |
| HUAWEI TECHNOLOGIES CO., LTD. | China | Technology | 3.1 | 42% | 50% | 0% | 0% | 0 | -176 | 0 | 0 | 7.602 | 11.846 |
| INDOFOOD SUKSES MAKMUR | Indonesia | Consumer Goods | 2.7 | 0% | 75% | 6% | 0% | 0 | -130 | 0 | 1 | 6.832 | 3.906 |
| INDORAMA VENTURES PUBLIC COMPANY LIMITED | Thailand | Basic Materials | 5.6 | 81% | 88% | 0% | 0% | 0 | -136 | 1 | 1 | 6.781 | 1.871 |
| INFOSYS LIMITED | India | Technology | 5.8 | 69% | 75% | 30% | 16% | 1 | -133 | 0 | 1 | 7.041 | 18.242 |
| JBS S.A. | Brazil | Consumer Goods | 3.1 | 35% | 56% | 2% | 0% | 0 | -104 | 0 | 1 | 7.492 | 3.146 |
| JOHNSON ELECTRIC HOLDINGS LIMITED | China | Industrials | 3.1 | 42% | 50% | 2% | 0% | 0 | -176 | 0 | 1 | 6.477 | 6.063 |
| KOC HOLDING A.S. | Turkey | Industrials | 4.6 | 62% | 75% | 0% | 0% | 0 | -151 | 0 | 1 | 7.420 | 7.289 |
| LARSEN & TOUBRO LIMITED | India | Industrials | 3.7 | 8% | 75% | 29% | 2% | 1 | -133 | 0 | 1 | 7.514 | 2,3455 |
| LATAM AIRLINES GROUP S.A. | Chile | Consumer Services | 4.5 | 73% | 56% | 7% | 3% | 1 | -31 | 0 | 1 | 7.285 | -1.595 |
| LDK SOLAR CO., LTD. | China | Oil, gas & energy | 1.3 | 19% | 19% | 0% | 0% | 0 | -176 | 1 | 1 | | |
| LENOVO GROUP LIMITED | China | Technology | 3.6 | 69% | 38% | 0% | 0% | 0 | -176 | 0 | 1 | 7.418 | 1.256 |
| LI & FUNG LIMITED | China | Consumer services | 3.9 | 65% | 50% | 1% | 0% | 0 | -176 | 0 | 1 | 6.914 | 5.599 |
| PUBLIC JOINT STOCK COMPANY OIL COMPANY LUKOIL | Russia | Oil, gas & energy | 2.2 | 46% | 19% | 0% | 0% | 0 | -148 | 1 | 1 | 7.884 | 7.073 |
| LUPIN LIMITED | India | Health care | 5.1 | 42% | 75% | 37% | 12% | 1 | -133 | 1 | 1 | 6.440 | 14.143 |
| CONTROLADORA MABE SA DE CV | Mexico | Consumer goods | 2.6 | 77% | 0% | 1% | 0% | 0 | -149 | 0 | 0 | | |
| MAGNESITA REFRATARIOS SA | Brazil | Basic materials | 2.9 | 54% | 31% | 1% | 0% | 0 | -104 | 1 | 1 | 6.318 | -8.875 |
| MAHINDRA & MAHINDRA LIMITED | India | Consumer goods | 6.7 | 85% | 75% | 40% | 26% | 1 | -133 | 0 | 1 | 7.197 | 3.138 |
| MARCOPOLO SA | Brazil | Industrials | 4.4 | 50% | 75% | 6% | 2% | 1 | -104 | 0 | 1 | 6.170 | 3.366 |
| MEXICHEM SAB DE CV | Mexico | Basic Materials | 4.6 | 77% | 56% | 4% | 2% | 1 | -149 | 1 | 1 | | |
| MTN GROUP LIMITED | South Africa | Telecommunication | 5.9 | 73% | 75% | 28% | 15% | 1 | -39 | 0 | 1 | 7.323 | 9.541 |
| NATURA COSMETICOS S.A. | Brazil | Consumer Goods | 4.7 | 65% | 75% | 0% | 0% | 0 | -104 | 0 | 1 | 6.408 | 7.822 |
| PUBLIC JOINT STOCK COMPANY MINING AND METALLURGICAL COMPANY NORILSK NICKEL | Russia | Basic Materials | 5 | 73% | 75% | 2% | 1% | 1 | -148 | 1 | 1 | 7.123 | 14.101 |
| PETROLIAM NASIONAL BERHAD | Malaysia | Oil, gas & energy | 6.3 | 88% | 100% | 2% | 2% | 1 | -146 | 1 | 0 | 8.164 | 4.557 |
| PTT PUBLIC COMPANY LIMITED | Thailand | Oil, gas & energy | 5.4 | 77% | 75% | 10% | 6% | 1 | -136 | 1 | 1 | 7.808 | 1.966 |
| HACI OMER SABANCI HOLDING ANONIM SIRKETI | Turkey | Industrials | 4.9 | 96% | 50% | 0% | 0% | 0 | -151 | 0 | 1 | | |

**Figure A1.** *Cont.*

| SAPPI LIMITED | South Africa | Basic materials | 3 | 58% | 25% | 6% | 1% | 1 | -39 | 1 | 1 | 6.715 | 2.935 |
|---|---|---|---|---|---|---|---|---|---|---|---|---|---|
| SASOL LIMITED | South Africa | Basic materials | 4.1 | 65% | 38% | 19% | 5% | 1 | -39 | 1 | 1 | 7.423 | 9.869 |
| PUBLIC JOINT STOCK COMPANY SEVERSTAL | Russia | Basic materials | 2.6 | 38% | 38% | 1% | 0% | 0 | -148 | 1 | 1 | 6.827 | -5.406 |
| SHANGHAI ELECTRIC (GROUP) CORPORATION | China | Industrials | 1.8 | 4% | 50% | 0% | 0% | 0 | -176 | 0 | 1 | 6.798 | 4.349 |
| SHUNFENG INTERNATIONAL CLEAN ENERGY LIMITED | China | Oil, gas & energy | 1 | 0% | 19% | 13% | 0% | 0 | -176 | 1 | 1 | 6.597 | 3.161 |
| SINOCHEM CORPORATION | China | Basic materials | 1.1 | 19% | 13% | 0% | 0% | 0 | -176 | 1 | 0 | | |
| SINOHYDRO CORPORATION LIMITED | China | Industrials | 2.4 | 65% | 6% | 0% | 0% | 0 | -176 | 0 | 0 | 6.699 | 0.917 |
| CHINA NATIONAL MACHINERY INDUSTRY CORPORATION | China | Industrials | 0.6 | 12% | 6% | 0% | 0% | 0 | -176 | 0 | 0 | 6.827 | 4.907 |
| SINOPEC ENGINEERING (GROUP) CO., LTD | China | Oil, gas & energy | 1.2 | 23% | 13% | 0% | 0% | 0 | -176 | 1 | 0 | 6.942 | 5.808 |
| SINOSTEEL GROUP CORPORATION-LIMITED | China | Basic materials | 0.8 | 12% | 13% | 0% | 0% | 0 | -176 | 1 | 0 | 6.674 | -1.510 |
| SUZLON ENERGY LIMITED | India | Oil, gas & energy | 5.8 | 65% | 75% | 32% | 16% | 1 | -133 | 1 | 1 | 6.396 | -18.649 |
| TATA CHEMICALS LIMITED | India | Basic Materials | 6.3 | 85% | 75% | 30% | 19% | 1 | -133 | 1 | 1 | 6.512 | 3.282 |
| TATA SONS LIMITED | India | Telecommunication | 7 | 88% | 75% | 45% | 30% | 1 | -133 | 0 | 1 | | |
| TATA CONSULTANCY SERVICES LIMITED | India | Technology | 6.5 | 88% | 75% | 30% | 20% | 1 | -133 | 0 | 1 | 7.110 | 26.358 |
| TATA GLOBAL BEVERAGES LIMITED | India | Consumer Goods | 6.5 | 88% | 75% | 31% | 20% | 1 | -133 | 0 | 1 | 6.172 | 2.997 |
| TATA MOTORS LIMITED | India | Consumer Goods | 6.5 | 88% | 75% | 31% | 20% | 1 | -133 | 0 | 1 | 7.590 | 5.507 |
| TATA STEEL LIMITED | India | Basic Materials | 6.4 | 88% | 75% | 30% | 20% | 1 | -133 | 1 | 1 | 7.417 | -1.374 |
| TENARIS S.A. | Argentina | Industrials | 3.9 | 73% | 44% | 0% | 0% | 0 | -54 | 0 | 1 | 7.196 | 3.240 |
| TENCENT HOLDINGS LIMITED | China | Telecommunication | 1.5 | 27% | 19% | 0% | 0% | 0 | -176 | 0 | 1 | 7.575 | 13,707 |
| THAI UNION GROUP PUBLIC COMPANY LIMITED | Thailand | Consumer goods | 5.5 | 88% | 75% | 1% | 1% | 1 | -136 | 0 | 1 | 6.518 | 4.584 |
| UNITED COMPANY RUSAL PLC, | Russia | Basic Materials | 3.3 | 65% | 31% | 4% | 1% | 1 | -148 | 1 | 1 | 7.124 | -12.134 |
| VEDANTA RESOURCES PLC | India | Basic Materials | 5.8 | 81% | 75% | 18% | 11% | 1 | -133 | 1 | 1 | 7.527 | -5.462 |
| VOTORANTIM GMBH | Brazil | Basic Materials | 3.8 | 69% | 38% | 8% | 2% | 1 | -104 | 1 | 0 | 6.351 | 44.464 |
| WANXIANG GROUP CORPORATION | China | Consumer goods | 0 | 0% | 0% | 0% | 0% | 0 | -176 | 0 | 0 | 6.779 | 1.691 |
| WEG S.A. | Brazil | Industrials | 3 | 54% | 38% | 0% | 0% | 0 | -104 | 0 | 1 | 6.607 | 8.105 |
| WIPRO LIMITED | India | Technology | 6.4 | 88% | 75% | 30% | 20% | 1 | -133 | 0 | 1 | 7.010 | 13.395 |
| YANZHOU COAL MINING CO., LTD. | China | Basic Materials | 2.1 | 8% | 50% | 5% | 0% | 0 | -176 | 1 | 1 | | |
| ZOOMLION HEAVY INDUSTRY SCIENCE AND TECHNOLOGY CO., LTD. | China | Industrials | 1.3 | 0% | 38% | 0% | 0% | 0 | -176 | 0 | 1 | 7.172 | 0.365 |
| ZTE CORPORATION | China | Technology | 5.9 | 88% | 88% | 0% | 0% | 0 | -176 | 0 | 1 | 7.270 | 2.649 |

**Figure A1.** Disaggregated data used for the companies in the sample.

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
