# Peer review of "Determinants of Corporate Anti-Corruption Disclosure: The Case of the Emerging Economics"

_sustainability, doi:10.3390/su13063462_

Round 1

Reviewer 1 Report

In the attached document, you can find my comments.

Author Response

Abstract:

I suggest including the implications of the findings and the main literature contribution of the manuscript.

Thank you very much for your suggestion. In the new version we have better defined the contribution in the abstract.

Introduction:

Review the English grammar because some sentences and paragraphs have been written in the present, past, and present perfect. There is inconsistency in the verb tenses in sentences.

 Thank you, we had sent this version to a professional editor to correct those aspects

Moreover, in the introduction part, authors need to include (1) the importance of the determinants of corporate anti-corruption disclosure, (2) findings of the research, (3) the main implications of the results, and (4) the division of the article. Include the importance to study this phenomenon in emerging economics. Remove some references from the introduction part and include them in the literature review and development of the hypothesis, especially, the last paragraph of the introduction part.

Thank you for your suggestion, is this new version we had add the importance of the determinant and define the different control variables. Try to explain better the finding of the research and the main implications of the results. Also as you recommended the have a rewrite the literature review of the introduction, and remove part of this literature review to the next section. Finaly we have done a introduction of the next section as you suggest us.

Literature review and development of hypothesis:

The authors need to discuss previous studies (empirical studies) and their results to develop the hypothesis. Moreover, re-structure the literature review for better understanding. For instance, authors might include theory and empirical studies (findings) that prove the positive relationship between variables, as well as, the negative relationship. Authors need to discuss the previous studies and their difference from their manuscripts.

Thank you for your recommendations, they had been very useful for us. In the present version, we have made substantial changes in this section; we propose the arguments of work done at the macro level from a political economy perspective that helps to see the relationship between the Press Freedom and the reduction of corruption in societies. We have tried to transfer this logic, corroborated by the literature, to the study of behavior at the micro or company level.

Empirical design:

 Include the period of the study. The equation is incorrect, authors need to add in the equation: time (t) and the error term. Also, include the Greek alphabet in notation (beta instead of B). Rewrite the methodology in a clear way for better understanding. Authors might add a table that includes the formulas or ranking to calculate dependent, independent, and control variables.

Thank you for your corrections, we have included the period of the study, corrected the equation, rewrited the methodology and added an explicative table (table 1) in the first paragraph of section 3. Materials and Methods

 The notation of the equation presented in the paper is an OLS regression, however, the authors mentioned that there is a logit regression. Review the formula for dependent variables.

Thank you, in the study we apply both methods to the same mathematical model and thats why we put it in the 3rd section. In addition, for clarity, in section 4 where we present the results, before referring to the results of the logit method, we have included the formulas used for the two models we have studied.

 In Table 3, the authors include model 1. However, in the empirical design (equations) there is not an equation that includes the only field.

Thank you for the point you make, we didn´t realize it. Now we have divided the empirical design into two models, we have added the missing equation. We have included it in the first paragraph after table 1

The conditions or assumptions of the selected case studies are not clearly explained. The authors suggest some considerations without any theoretical or empirical background. The qualitative and quantitative approaches need to include more arguments for the research design.

Mention the method of collecting data.

Thank you for your suggestion. We have explained the way to select case studies better as we have rewroten the methodology clearer and we have add table 1 for clarity.

 Results:

Authors need to improve the presentation of tables. Include the same number of decimals in all tables.

In Tables 3 and 4, authors need to include the value of t-statistics. Authors need to include a note with the meaning of stars * (significance level). Thank you for all of your corrections, we correct the tables and include 3 decimals in all the tables.

Thank you. We now include in table 4 (previously 3) the value of wald and in table 5 (previously 4) t-stadistic, and the significance levels after the tables.

Authors need to analyze their findings and their implications in the financial, economic, and social behavior of the company.

Thank you, we try to do this in our new version, and we have also added in our limitations and future lines of research the importance of carrying out this analysis in greater detail.

Include the similarities/differences between authors’ results with prior findings.

Thank you, we think this is already done. Just before and after the table 3 those differences and similarities are explained.

Include more references, such as:

  • • Tulcanaza-Prieto, H. Shin, Y. Lee, and C. Lee, “Relationship among CSR Initiatives and Financial and Non-Financial Corporate Performance in the Ecuadorian Banking Environment,” Sustainability, vol. 12, no. 1621, pp. 1–17, 2020.
  • • Tulcanaza-Prieto, Lee, Y., and J. Koo, "Leverage, Corporate Governance, and Real Earnings Management: Evidence from Korean Market," Global Business & Finance Review, vol. 25, no. 4, pp. 51–72, 2020.

Thank you for your suggestion, we have included the first reference suggested by you, we think the second one does not fix so well to our paper, and taking into consideration we already have a huge number of references we decided not to include it.

Explain clearly the differences between Table 3 and Table 4 because they look similar, but with different results.

Thank you, we explained in the paper the reason to include both tables for more evidence of the robustness we ran a lineal regression. The results of the lineal regression analysis (Table 4) remain similar to the results obtained on the logit regression analysis”

 Conclusions:

State the main findings of the research, the contribution of the article, the limitations of the research, and future studies.

 Thank you very much for these recommendations. In the present version and taking into account your suggestions in the previous sections, we have incorporated those aspects that were absent, and we believe that they now improve the process of discussing the results and provide more concise conclusions that are better aligned with the framework used in the study.

References are huge, please review them and only use papers that contribute to the article.

Thanks again, we have also make a critical reading of our references and try to include only the references that directly and significantly contribute to our paper.

Reviewer 2 Report

This paper provides novel insight into a highly-interesting and highly-challenging issue, namely corporate corruption disclosure. The study is interesting and informative, although the resulting manuscript needs improvements and additions.

GENERAL RECOMMENDATIONS

  • You need to mention that approaches of corruption disclosure can be very different. E.g., corporate codes of conduct, corporate web-pages, etc are full of the relevant information. The both real cases of corruption, as well as anti-corruption policy can be disclosed. So, the issue is very wide.
  • You may note that corruption can be measured very differently. In some countries, it appears too high only because of strict legal regulations, where all what is not prescribed is corruption. This is not the case in the other countries. More generally: what is corruption in one country is not so in the other. Probably, you need to take into account the factor of administrative discretion (see the works by Prof. J. Vaughn and T. Hutzschenreuter). I feel that when this discretion is broad, corruption shrinks "automatically" (this seems to be to case of the USA).
  • Note, please, that the relationship between corruption and innovation activity is not so simple. Sometimes, corruption, which is bad ethically, provides good mechanisms for innovation growth. You may try to find the relevant works for citation (but I do not insist on doing this).
  • You should note a difference of the influence of the mass media (press) between the countries and their readability. In some countries, people simply do not trust newspapers (both state-controlled and independent) or do not read them on regular basis. In this case, press freedom means almost nothing.
  • I encourage you to present your results also by countries, as well as to relate them to the size of the national economies.

OTHER RECOMEMNDATIONS

  • All statistical tools should be named in the methodological section.
  • If you do not separate your conclusions, the last section should be titled "Discussion and Conclusion". Please, consider the limitations of your study there.
  • To me, it would be challenging to judge China or Russia as emerging economies.
  • The writing is ok, but, please, avoid too short paragraphs and half-finished phrases.
  • Do not forget, please, to check the language once again. Small errors may occur here and there.

Author Response

Comments and Suggestions for Authors

This paper provides novel insight into a highly-interesting and highly-challenging issue, namely corporate corruption disclosure. The study is interesting and informative, although the resulting manuscript needs improvements and additions.

Thank you very much for your kind comments

GENERAL RECOMMENDATIONS

You need to mention that approaches of corruption disclosure can be very different. E.g., corporate codes of conduct, corporate web-pages, etc are full of the relevant information. The both real cases of corruption, as well as anti-corruption policy can be disclosed. So, the issue is very wide.

Thank you for your suggestion, we have included in the second paragraph a sentence indicating the existence of various ways to disclose anti-corruption information.

 You may note that corruption can be measured very differently. In some countries, it appears too high only because of strict legal regulations, where all what is not prescribed is corruption. This is not the case in the other countries. More generally: what is corruption in one country is not so in the other. Probably, you need to take into account the factor of administrative discretion (see the works by Prof. J. Vaughn and T. Hutzschenreuter). I feel that when this discretion is broad, corruption shrinks "automatically" (this seems to be to case of the USA).

 Note, please, that the relationship between corruption and innovation activity is not so simple. Sometimes, corruption, which is bad ethically, provides good mechanisms for innovation growth. You may try to find the relevant works for citation (but I do not insist on doing this).

 You should note a difference of the influence of the mass media (press) between the countries and their readability. In some countries, people simply do not trust newspapers (both state-controlled and independent) or do not read them on regular basis. In this case, press freedom means almost nothing.

I encourage you to present your results also by countries, as well as to relate them to the size of the national economies.

Thank you for your suggestions, we think all of them are really interesting so we refer to them in the paper and take note of them to include all these aspects in our future researches.

 OTHER RECOMEMNDATIONS

 All statistical tools should be named in the methodological section.

 If you do not separate your conclusions, the last section should be titled "Discussion and Conclusion". Please, consider the limitations of your study there.

 To me, it would be challenging to judge China or Russia as emerging economies.

 The writing is ok, but, please, avoid too short paragraphs and half-finished phrases.

Do not forget, please, to check the language once again. Small errors may occur here and there.

Thank you for your invaluable recommendations, we made some changes in method and materials section, to explain better all statistical tools we used. We change the title to the discussion section and we have included the limitation of the study.

I´m agree with you in the challenge to name China and Russia as emerging economies nowadays, but we consider the rapid evolution of their economies doesn´t allow them to change their political structure at the same pace. We focused in the political structure that underlies these countries, more than in their current economic potential.

Finally, we had sent this version to a professional editor to correct language related issues, and check our writing.

Round 2

Reviewer 1 Report

The revised version contains the main suggestions.

Congratulations.

Author Response

Thank you very much for all your suggestions, it has been an invaluable help in improving our work.

Reviewer 2 Report

I'm very satisfied with this revision. I see only one issue to improve (this is not too difficult). I see a lot of methodological notions in Results. These MUST be moved to the methodological section.

Author Response

I'm very satisfied with this revision. I see only one issue to improve (this is not too difficult). I see a lot of methodological notions in Results. These MUST be moved to the methodological section.

Thank you very much for all the suggestions you have made, it has been an invaluable help in improving our paper.
In response to your request, we have tried to upload all the methodological notions to the Materials and methodology section: the logit formulas and the justification for using Spearman's correlations.